# Effect of Mg on the Formation of Periodic Layered Structure during Double Batch Hot Dip Process in Zn-Al Bath

**DOI:** 10.3390/ma14051259

**Published:** 2021-03-06

**Authors:** Mariola Saternus, Henryk Kania

**Affiliations:** 1Department of Metallurgy and Recycling, Faculty of Engineering Materials, Silesian University of Technology, Krasińskiego 8, 40-019 Katowice, Poland; 2Department of Advanced Materials and Technology, Faculty of Engineering Materials, Silesian University of Technology, Krasińskiego 8, 40-019 Katowice, Poland; henryk.kania@polsl.pl

**Keywords:** hot dip galvanizing, Zn-AlMg bath, periodic layered structure

## Abstract

The article presents the results of studies on the influence of Mg on the formation of the periodic layered structure of the Zn-AlMg coatings. These coatings were produced by the double batch hot dip method in a Zn bath and then in a Zn-Al(Mg) bath with a content of 15, 23, 31 wt.% Al and 3, 6 wt.% Mg. The microstructure of the coatings (OM, SEM) was revealed and the phase composition (XRD) obtained in two-component Zn-Al baths and Zn-AlMg baths were determined. The periodic layered structure was found to consist of alternating FeAl_3_ phase layers and a bath alloy (Zn + Al + Mg). Moreover, it was observed that the addition of 3 wt.% Mg reduces the thickness of the coating in baths containing 23 and 31 wt.% Al. However, the addition of 6 wt.% Mg causes complete disappearance of periodic layered structure in a bath with 23 wt.% Al. In a bath with a content of 31 wt.% Al the addition of 6 wt.% Mg creates a compact layer consisting of the FeAl_3_ phase containing the precipitation of the MgZn_2_ phase and Fe_2_Al_5_ phase. Such a structure of the coating transition layer limits the growth of the periodic layered structure in the coating.

## 1. Introduction

The hot dip galvanizing process is an efficient and economical way to protect steel structures against corrosion. Zinc coatings have many advantages; they provide barrier and sacrificial protection of the steel surface and are resistant to mechanical damage [1]. As a result, they have found new applications, e.g., for covering wires intended for drawing at high speeds [2], as well as for the protection of alloy steel products after thermal improvement [3]. However, the increasingly higher requirements for the corrosion resistance of steel products mean that the zinc coating often does not meet these expectations. Additionally, the widespread use of Pb [4], Bi [5] and Sn [6] alloying additions may lead to a reduction in their corrosion resistance [7]. Therefore, new zinc baths based on Zn-Al-Mg alloys for hot dip galvanizing have been developed. Coatings obtained in Zn-Al-Mg baths by the continuous method on sheets show much better corrosion resistance and susceptibility to plastic processing [8]. In batch hot dip methods, there is a need for fluxing and a much longer bath time and higher temperature. Under such conditions, the reactions between the iron and the bath are very violent, leading to the formation of complex coatings. Wang [9], conducting research in the Zn-45Al-Mg bath, found a periodic layered structure in the coating. Similarly, Gao et al. [10] show the formation of a periodic layered structure in baths with a content of 15–55 wt.% Al. Wang [9] claims that the presence of a periodic layered structure has a positive effect on the corrosion resistance of coatings. However, it seems that due to the significant increase in coating thickness, the presence of a periodic layered structure is not justified. According to Gao et al. [10] periodic layered structure formation can be inhibited by the addition of Mg to the bath.

If, as a result of the reaction between the components of the system, more than one layer of the same product is formed, a regularly repeating system of reaction products is formed in the reaction zone [11]. This phenomenon was first observed by Osinski et al. in the contact of the Fe-15 wt.% Si alloy with a zinc bath [12]. Liu et al. [13] also found the formation of a periodic layered structure during the reaction of FeCr alloys with liquid zinc. Other studies have focused on the formation of a periodic layered structure in the solid state, including zinc (Zn/Ni3Si [14], Zn/Cu_x_Ti_y_ [15], Zn/Co_2_Si [16]), aluminum (Al/U10Mo [17], Al/(Ni,W) [18]), and magnesium (Mg/SiO_2_ [19]) systems. The formation of a periodic layered structure during the hot dip galvanizing process in Zn-Al and Zn-AlMg baths, however, remains unclear.

Due to the difficulty of producing continuous coatings in Zn-Al-Mg baths by batch hot dip methods, so far research on shaping the coating structure has been carried out at very high temperatures (590–610 °C) [10,20,21]. High bath temperatures increase the tendency to form a periodic layered structure. The use of the double batch hot dip method for the production of coatings allows us to lower the bath temperature while ensuring the processability and continuity of the obtained coatings. The study investigated the effect of Mg addition to the Zn-Al baths on the formation of periodic layered structure in coatings obtained by double batch hot dip methods.

## 2. Experimental

### 2.1. Preparation of Coating

S235JRG2 steel was selected for the tests with content of 0.138 wt.% C, 0.021 wt.% Si, 0.743 wt.% Mn, 0.0086 wt.% S and 0.0088 wt.% P. The coatings were prepared by the double batch hot dip method on steel samples with dimensions of 50 × 25 × 2 mm. First, the samples were immersed in a Zn bath at 450 °C for 30 s. After removal from the Zn bath, the samples were immersed in the Zn-Al or Zn-AlMg bath for 30, 60, 120 and 240 s. The Al content in the bath was 15, 23 and 31 wt.%, while the content of Mg was 3 and 6 wt.%. The charge materials for the bath were Super High Grade Zinc, ZnAl4 alloy and AlMg25 alloy. After removing from the Zn-Al bath, the samples were cooled in two variants: in air and water, while the samples from the Zn-AlMg bath were cooled only with air. The temperature of the ZnAl and Zn-AlMg baths depended on the Al content. It was assumed that the hot dip process would be carried out at the lowest possible temperature, which for technological reasons is determined as at least 20 °C higher than the melt solidification point. Based on the earlier studies of the solidification point of Zn-Al and Zn-AlMg alloys [19], it was assumed that the bath temperature containing 15 wt.% would be 460 °C of a bath containing 23 wt.%—510 °C and a bath containing 31 wt.%—560 °C.

Before immersion in the Zn bath, the steel samples were degreased in an acidic HydronetBase solution for 5 min, then etched in a 12% HCl solution for 10 min, rinsed in water and fluxed in a TegoFlux60 solution for 2 min. Such prepared samples were dried at 120 °C for 15 min.

### 2.2. Characterization Methods

The optical microscope Olympus-GX51 (OM) (Olympus Corporation, Tokyo, Japan) and the scanning electron microscopy (SEM) Hitachi S-3400 N (Hitachi, Tokyo, Japan) equipped with an energy dispersion spectroscope (EDS) were used to analyze the microstructure and investigate the microchemical composition. The analySIS software Olysia m3 (Olympus Corporation, Tokyo, Japan) was used for image recording (OM) and coating thickness measurement. The coating thickness was measured in 10 randomly selected points on a metallographic specimen on 3 samples of the same type. Noran Instruments—System Six software (Thermo Fisher Scientific, Waltham, MA, USA) was used to record the microstructure image (SEM) and to study chemical composition in microregions (EDS). 

The X-ray phase analysis was carried out on a X’Pert 3 X-ray diffractometer (Malvern Panalytical, Malvern, UK), using a lamp with a copper anode (λCuKα = 1.54178 Ǻ) with a current of 30 mA at a voltage of 40 kV, and a graphite monochromator. The recording was made continuously with a step of 0.026° in the range from 10 to 90° 2θ. The tests were carried out in the characteristic areas of the surface coating after grinding the top of the coating. The depth of grinding the coating was determined by observing the cross-section of the coating under a light microscope. The phase composition was identified using the ICDD PDF-2 and COD Database.

## 3. Results and Discussion

### 3.1. Microstructure, Kinetics Growth and Phase Constituent of Binary Zn-Al Coatings

Figure 1 shows the microstructure of the coatings obtained after the immersion of the steel samples in a two-component Zn-Al bath with different Al content. The coating obtained in a bath with a content of 15 wt.% Al (Figure 1a) shows a two-layer structure. A compact layer is formed at the substrate, which forms the transition layer of the coating. The layer increases with immersion in the bath. The compact layer has a considerable thickness and its morphology is typical for the phases of the Fe-Al system growing on a steel substrate in Zn-Al baths [20]. The compact layer is covered with an outer layer, which is formed as a result of pulling the Zn-Al bath alloy and its crystallization on the sample surface [22]. 

The structure of the coating changes as the Al content in the bath increases. In the coating obtained in the bath with a content of 23 wt.% (Figure 1b) initially a compact layer formed on a steel substrate turns into a periodic layered structure. The transition layer thus formed is covered with an outer layer. The periodic layered structure consists of alternating light layers with Fe-Al phase morphology and dark gray layers with a solidified Zn-Al bath morphology. These layers have an orientation approximately parallel to the steel substrate. 

In order to confirm the presence of phases in the coating, phase composition studies (XRD) were carried out. Near the coating surface (approximately the area corresponding to the line (1) in the cross-section of the coating in Figure 2b) the presence of Al, Zn and the FeAl_3_ intermetallic phase (Figure 2a) was found. After grinding the initial approximately 260 μm of the coating (approximately on the surface corresponding to line (2) in the cross-section of the coating in Figure 2b), the presence of Al, Zn and FeAl_3_ was also identified. However, after grinding another approximately 260 μm deep into the coating (approximately on the surface corresponding to the line (3) on the cross-section of the coating in Figure 2b), the presence of similar phases was found: Al, Zn and FeAl_3_, but also the presence of Fe. On this basis, it can be concluded that the periodic layered structure consists of alternating layers of the FeAl_3_ intermetallic phase and layers of a solidified Zn + Al bath. On the other hand, the compact layer is made of the FeAl_3_ intermetallic phase. The presence of Fe in this region confirms that the test area covered the substrate–coating interface. No Fe_2_Al_5_ phase was found in this area. Many authors argue that in the layer formed on the iron surface in the Zn bath, containing Al, the Fe_2_Al_5_ phase is the major component of the transition layer. Detailed studies carried out by McDevitt et al. [23] using the X-ray diffraction method (XRD) and transmission electron microscopy (TEM) confirm the presence of both Fe_2_Al_5_ and FeAl_3_ phases. They claim that the FeAl_3_ phase is formed first. Then the Fe_2_Al_5_ phase grows at the expense of the FeAl_3_ phase located between the substrate and the FeAl_3_ phase layer. According to McDevitt et al. [23], XRD methods are unable to detect FeAl_3_ phase, which may lead to misinterpretation of test results. However, McDevitt investigated the reactions taking place directly at the interface Fe substrate/Zn-Al bath in a bath with Al content corresponding to a bath for continuous galvanization (0.2–0.25 wt.% Al). Horstmann [24] claims that with a higher Al content in the bath, the FeAl_3_ phase may be the first to form on the iron surface, which then changes into the Fe_2_Al_5_ phase. Ghuman and Goldstein [25] indicate the presence of the Fe_2_(AlZn)_5_ phase during the reaction with the bath containing less than 1% Al. On the other hand, if zinc contains from 5% to 10% Al, the triple phase changes to Fe(ZnAl)_3_. However, Chen et al. [20] found the presence of the Fe_2_Al_5_Zn_x_ phase located at the boundary with the substrate and the FeAl_3_Zn_x_ phase located in the upper zone of the layer in eutectic baths.

Increasing the Al content in the bath to 31 wt.% changes the structure. The coating is made of a compact layer on a steel substrate, which then transforms into a porous structure (Figure 1d). This layer is more compact in the area of the inner shell and more porous on the outside. The appearance of the cross-section indicates the presence of the Fe-Al intermetallic phases in the coating [20]. The characteristic porous structure is formed in the coating during slow cooling in the air. The coating cooled after leaving the bath at high speed in water shows no porosity (Figure 1e). It shows a distinct periodic layered structure with a thin, compact layer on the substrate and an outer layer on the surface. The morphology of the structural components indicates the occurrence of alternating layers of the intermetallic phase of the Fe-Al system and the layers of the bath alloy (Zn + Al), which show a eutectoid structure. Therefore, it should be stated that the porous structure is formed during the slow cooling in air. After removing the sample from the bath, the areas in the coating filled with the bath due to the lower liquidus temperature [26] remaining liquid until freezing point is reached. This proves that the reaction between Fe and the drawn bath still takes place, and its course is additionally facilitated by the heat generated as a result of the endothermic reaction of Fe with Al [27]. The violent course of the reaction during the slow cooling of the coating in air leads to the almost complete exhaustion of the bath and the formation of porosity in the coating. Such a rapid course of the reaction was not observed in the bath containing 23 wt.% Al (Figure 1c). It should be noted, however, that the Al content in the bath determines the minimum temperature at which the hot dip process can be carried out. The formation of the porous structure in the Zn-31Al bath is therefore also influenced by the higher reaction temperature.

The formation of the periodic layered structure intensely affects the thickness of the coatings. The growth kinetics of the coatings obtained in the Zn-23Al and Zn-31Al baths are shown in Figure 3. As the thickness of the outer layer is small, changes in the thickness of the entire coating over time characterize the kinetics of the coating layer growth relatively accurately. The determined trend line and a high correlation coefficient (shown in Figure 3) indicate a linear course of the kinetics of coating growth, both in the bath with 23% and 31% Al. The thickness increase of the coatings is very rapid. After immersion of 240 s, a coating with an average thickness of 560.25 ± 9.63 µm in the Zn-23Al bath and 1694.50 ± 124.62 µm in the Zn-31Al bath was obtained, respectively. Cooling the coating in water after taking it out of the bath does not significantly affect the thickness of the coating obtained in the Zn-23Al bath. On the other hand, in the Zn-31Al bath, cooling in water causes a significant reduction in the thickness of the coating. After 240 s, the average thickness of the coating was 1498.66 ± 19.03 µm.

### 3.2. Effect of Magnesium

#### 3.2.1. Cross-Section of Coatings

The structure of coatings obtained in Zn-23Al baths with 3 wt.% and 6 wt.% Mg at the immersion time of 120 s and 240 s and the temperature of 510 °C is shown in Figure 4. At the content of 3 wt.% Mg in the bath after the immersion time of 120 s (Figure 4a) compared to the structure of the coating obtained in the bath with the same Al content without the addition of Mg (Figure 1b), the transition layer of the coating has a lower thickness and is more compact. In the periodic layered structure area, there is only one pair of layers, the intermetallic phase of the Fe-Al system/bath alloy layer (Zn + Al + Mg). It can be assumed that this is the initial stage of creating a periodic layered structure. Extending the immersion time to 240 s resulted in a clear formation of a periodic layered structure (Figure 4b). However, a maximum of 2–3 pairs of alternating intermetallic layers—(Zn + Al + Mg) can be observed. With the same process parameters, the coating obtained in the bath without the addition of Mg showed approximately 6–7 alternating layers (Figure 2b). Higher Mg addition—6 wt.% to Zn-23Al bath causes complete inhibition of periodic layered structure formation. In the structure of the coating obtained in the Zn-23Al6Mg bath, a compact layer of relatively small thickness and an outer layer (Zn + Al + Mg) are formed during immersion of 120 and 240 s (Figure 4c,d).

Figure 5 shows the structure of coatings obtained in the Zn-31Al6Mg bath for various immersion times at the temperature of 560 °C. Already at the shortest immersion time of 30 s, the initial stage of periodic layered structure formation was observed (Figure 5a). With the increase of the immersion time up to 60 s, the formation of a periodic layered structure was observed, consisting of several alternating intermetallic—(Zn + Al + Mg) layers (Figure 5b). As the dip time is lengthened to 120 s, the number of pairs of layers continues to increase as shown in Figure 5c. At the same time, the compact layer starts to grow on the ground side, which shows two distinct zones. Zone I is thicker and more heterogeneous in structure, while Zone II is thinner and more compact. Further extension of the immersion time to 240 s increases the thickness of zone I, while the thickness of zone II does not change significantly (Figure 5d).

Based on the presented research results, it can be assumed that with the immersion time of 120 and 240 s, the formation of periodic layered structure disappears in the Zn-23Al6Mg bath, while its significant limitation is observed in the Zn-31Al6Mg bath. At 3 wt.% Mg was also found to slow down the formation of periodic layered structure in the Zn-23Al bath. However, the tests conducted in the Zn-31Al bath containing 3 wt.% Mg at 540 °C [28] showed no inhibition of periodic layered structure formation. Gao et al. [10] argue that the periodic layered structure is completely destroyed with longer immersion times. In the Zn-45Al0.5Mg bath at the temperature of 590 °C, they found the disappearance of the periodic layered structure on the surface of the coating after the immersion time of 120 s and its transformation into a more compact structure of the FeAl_3_Zn_x_ phase. On the other hand, in the Zn-55Al0.5Mg bath at the temperature of 610 °C, a similar effect was observed after 80 s. Gao et al. [10], referring to Chen et al. [20] and Nishimoto et al. [21] explain this by the fact that when the immersion time is too long, the growth rate of the intermetallic phases of the Fe-Al system decreases due to the attack of the liquid phase coming from the bath. While the bath attack on the Fe-Al system phase layers can explain their disappearance on the coating surface as a result of the dissolution processes taking place, it does not explain the formation of a more compact structure of the FeAl_3_Zn_x_ phase in this area. Gao et al. [10] also indicate that some incubation time is required to initiate the growth of the periodic layered structure. In the Zn-45Al0.5Mg bath at the temperature of 590 °C, the increase in periodic layered structure was observed after the immersion time of 20 s, while in the Zn-55Al0.5Mg (610 °C) bath after 10 s. The study of the kinetics of periodic layered structure formation in the Fe30Cr70/Zn system conducted by Liu et al. [13] also showed that their formation requires incubation time. A similar phenomenon was observed in the structure of the tested coatings. At the 30 s immersion time in the Zn-23Al bath (Figure 6a) and after the 60 s immersion time in the Zn-23Al3Mg bath (Figure 6b), no periodic layered structure was observed and only a compact layer was formed on the steel surface. Likewise in the Zn-23Al3Mg bath, the initiation of periodic layered structure formation observed after 120 s (Figure 4a) confirms that its formation requires an incubation time, which depends on the bath composition and process temperature.

#### 3.2.2. Coating Thickness

The relationship between the average thickness of the coatings obtained in the Zn-23Al and Zn-31Al baths with the addition of 3 and 6 wt.% for the immersion time from 30 to 240 s is shown in Figure 7. In the Zn-23Al bath with the content of 3 wt.% Mg, determined at 510 °C, the trend line shows a linear increase in coating thickness (Figure 7a). However, Mg reduces the thickness of the coating. After 240 s, the obtained coating thickness was 413.62 ± 15.30 µm, which is 146 µm less than the thickness of the coating obtained in the two-component Zn-23Al bath. Increasing the Mg content in this bath to 6 wt.% causes a rapid reduction in the thickness of the coating. After 240 s, a coating with an average thickness of 91.42 ± 10.44 µm was obtained. The compact layer (Figure 4c,d) shows a slight variation in thickness. The growth kinetics of Fe-Al intermetallic layer is described by the equation [29]:*y* = *k**t^n^*,(1)
where: *y*—thickness of intermetallic layer [μm], *k*—growth rate constans [μm/s^n^], *t*—growth time [s], *n*—growth rate index.

In the equation of the kinetics of compact layer growth (Figure 7b), the growth rate index n is 0.51. The growth rate index n close to 0.5 indicates that the growth of the compact layer is controlled by diffusion [29]. The addition of 6 wt.% to Zn-23Al bath changes the growth mechanism of the Fe-Al intermetallic layer. The periodic formation of a layered structure was not observed in this bath.

Similar changes in the coating growth kinetics did not occur after adding Mg to the Zn-31Al bath (Figure 7c). At the addition of 3 and 6 wt.% Mg, the development of the periodic layered structure in the coating was observed, which makes the coating growth kinetics linear. However, the addition of Mg reduces the thickness of the coating. With a content of 3 and 6 wt.% Mg in the Zn-31Al bath, after the immersion time of 240 s, coatings with a thickness of 872.25 ± 17.98 μm and 858.32 ± 9.52 μm were obtained, respectively. In the Zn-31Al two-component bath, the coating thickness was approximately two times greater.

#### 3.2.3. Microstructure (SEM) and EDS Analysis

The addition of Mg to the Zn-Al bath reduces the thickness of the coating and limits the growth of the periodic layered structure. This is especially true of coatings obtained in baths containing 6 wt.% Mg. The growth of periodic layered structure was completely lost in Zn-23Al6Mg bath. On the other hand, in the Zn-31Al6Mg bath, a significant change in its thickness was observed with the simultaneous formation of a thicker compact layer consisting of two zones.

The SEM image of the coatings obtained in the Zn-31Al6Mg bath is shown in Figure 8. The results of the chemical composition tests in EDS microareas are presented in Table 1. Fe-Al intermetallic layers can be observed in the area of periodic layered structure, in which 56.9 at.% Al, 18.9 at.% Fe, 22.3 at.% Zn, 1.9 at.% Mg (Figure 8b, Table 1, point 2) were found. The atomic fraction of Fe to Al is close to the 1:3 ratio with high accuracy. This proportion of components is appropriate for the FeAl_3_ phase. The structure of the alloy layer (Zn + Al + Mg) is more complex. In Al there are dendrites from the Zn solution (Figure 8b, Table 1, point 1) with the content of 85.6 at.% Al and 14.4 at.% Zn. The interdendritic spaces are rich in Zn and Mg (Figure 8b, Table 1, point 3), which may indicate the presence of the Zn-Mg phase. There were also fine particles of Zn-Al solution in their area (Figure 8b, point 4). There are also Zn-rich regions in the alloy layer (Zn + Al + Mg). In the darker, heterogeneous region at point 5 (Figure 8c), 89.8 at.% Zn and small amounts of Al and Mg are found, while in the brighter area with a more compact structure in point 6 there is a content of 58.1 at.% Zn and 41.9 at.% Al. The chemical composition of these regions indicates the presence of Al solution in Zn.

The compact layer of the coating obtained in the Zn-31Al6Mg bath is shown in Figure 8d. The structure of zone I is heterogeneous. In the darker area in point 7 (Table 1), the presence of Al, Fe and Zn can be found. The Fe to Al ratio is close to the 1:3, which may indicate the presence of the FeAl_3_ phase in this area. The high concentration of Zn and Mg in the brighter region at point 8 shows that this is the phase of the Zn-Mg system. The structure of zone II is more compact. There is a high concentration of Al and Fe in this zone. The share of Fe to Al at points 9 and 10 is close to 2:5, which indicates the presence of the Fe_2_Al_5_ phase. Additionally, it can be stated that no Zn was found in the chemical composition in the area closer to the ground (point 10). However, a high concentration of zinc occurs in point 9.

#### 3.2.4. X-ray Phase Analysis (XRD)

The phase composition of the coating was additionally confirmed by the XRD method, as shown in Figure 9. After grinding the initial approximately 100 μm of the coating (approximately on the surface corresponding to the line (1) in the coating cross-section in Figure 9b), the presence of Al, Zn and the intermetallic phases FeAl_3_ and MgZn_2_ was identified in the periodic layered structure zone. After grinding approximately 560 μm from the coating surface (approximately on the surface corresponding to the line (2) in the coating cross-section in Figure 9b), the presence of Al, Zn, FeAl_3_ and MgZn_2_ was also found. However, after grinding approximately 880 μm into the coating (approximately on the surface corresponding to the line (3) on the cross-section of the coating in Figure 9b), the presence of Al, Zn, FeAl_3_ intermetallic phase was found, and the presence of intermetallic phase Fe_2_Al_5_ and Fe was also confirmed. The cross-section (3), however, did not confirm the presence of the MgZn_2_ intermetallic phase.

Based on the SEM-EDS and XRD analysis, the structure of the coating obtained in the Zn-31Al6Mg bath can be determined. The periodic layered structure is made of alternating FeAl_3_ phase layers and a bath alloy (Zn + Al + Mg). The alloy layers (Zn + Al + Mg) are composed of dendrites of Zn solid solution in Al. The interdendritic spaces are filled mainly with the MgZn_2_ intermetallic phase and small areas of Al in Zn solution. The compact layer consists of two zones. Zone I is composed of the FeAl_3_ phase, in which the MgZn_2_ phase regions are located. The structure of the FeAl_3_ phase is heterogeneous and areas of the Al and Zn solutions may also be located there. A layer of Fe_2_Al_5_ is formed directly on the steel substrate in zone II.

In research by Ranjan et al. [30] carried out in the Zn-21.1Al bath containing the Bi, RE or Si additives, the presence of the Fe_2_Al_5_ phase containing dissolved Zn and Si was found [30]. The presence of the Fe_2_Al_5_ phase at the border with the substrate was also observed by Gao et al. [10]. The coating obtained in the Zn-55Al-0.5Mg bath was built on a steel substrate of a thin layer of the zinc-dissolving Fe_2_Al_5_ phase, which then transformed into a periodic layered structure composed of alternating layers of the FeAl_3_ phase and alloy (Al + Zn). However, this coating was obtained at a higher temperature of 610 °C. According to Peng et al. [31] during the reaction of Fe with the Zn-Al bath, the intermetallic phases FeAl_3_ and Fe_2_Al_5_ are formed, which contain dissolved zinc. Perrot et al. [32] observed that the FeAl_3_ phase is formed in the initial growth phase, which then transforms into the Fe_2_Al_5_ phase. Ranjan et al. [30] note, however, that based on the analysis of the equilibrium systems Fe-Al [26] and Fe-Al-Zn [33], the Fe_2_Al_5_ intermetallic phase has the highest liquidus temperature. Therefore, nucleation of this phase is most likely to occur first. According to the Fe-Al-Zn equilibrium system, the Fe_2_Al_5_ phase is stable when the content of 0.133 wt.% Al in liquid zinc is exceeded [34]. Under these conditions, the solubility of Zn in the Fe_2_Al_5_ phase may reach 22 wt.% [35]. Ranjan et al. [30] did not observe the FeAl_3_ phase at the interface, which is based on the claim that the Fe_2_Al_5_ phase is the first to form on a steel substrate.

The conducted tests confirmed that the addition of Mg reduces the thickness of the coating and limits the formation of a periodic layered structure. Mg is located in the coating in the form of the MgZn_2_ intermetallic phase. Gao et al. [10] conducting research in baths with a content of 15–55 wt.% Al and 0.5–3.0 wt.% Mg, claim that Mg does not participate in the formation of intermetallic phases of the Fe-Al system and is not present in the coating in the form of the Zn-Mg system phase only in elemental form, whereas Xie et al. [36] showed the formation of the MgZn_2_ intermetallic phase in coatings obtained in eutectic ZnAl baths containing more than 3 wt.% Mg. Similarly, Liu et al. [13] claim that the MgZn_2_ phase appears at higher Mg contents in the bath. According to research by Xie et al. [36] Mg segregates at the grain boundaries of the Fe-Al system phases, which slows down the growth kinetics of these phases. The influence of Mg on the formation of the periodic layered structure has not yet been fully explained. In the two-component Zn-Al baths, the formation of a periodic layered structure is explained by the occurrence of stresses. Chen et al. [20] show that during the reaction in Zn-Al baths, the Fe_2_Al_5_Zn_x_ phase is formed on the steel surface in very short times, which is covered with a FeAl_3_Zn_x_ phase layer. The formation of the Fe_2_Al_5_Zn_x_ phase is associated with a significant increase in volume, which causes stresses in the area of the substrate/coating interface, causing cracking and detachment of this phase layer during its growth [37]. The spalling effect, consisting in delamination of the diffusion layer, is also known in aluminum coatings. This phenomenon is explained by the presence of stresses generated in the diffusion layer, which eventually cause delamination [38]. Selverian et al. [37] observed the formation of layers of different morphology in baths with high concentrations of Al (45–75% Al), which are detached as continuous FeAl_3_ phase layers with a bent shape or separate particles. The bending of the FeAl_3_ phase layer explains the stress occurrence at the Fe_2_Al_5_/FeAl_3_ interface. On the basis of the structural tests of coatings obtained in the Zn-31AlMg6 bath, it can be stated that the stabilization of the compact layer growth occurs after the formation of zone I in the coating, which is made of FeAl_3_ phase containing MgZn_2_ phase precipitates. This leads to an inhibition of the growth of the periodic layered structure and creates the conditions for growth on the substrate of the Fe_2_Al_5_ phase layer. However, no Fe_2_Al_5_ phase was found at the time of periodic layered structure formation. Analysing the results of the research and relying on the results of the research by McDevitt et al. [23] and Horstmann [24], it can be assumed that the first FeAl_3_ phase is formed on the substrate, which increases reaching the thickness at which it transforms into the Fe_2_Al_5_ phase at the border with the substrate. This may be the reason for stress formation and the delamination of the FeAl_3_ phase layer. The studies of the structure of the coatings obtained in the Zn-31Al6Mg bath show that the FeAl_3_ phase containing MgZn_2_ precipitations does not show a tendency to delamination. Building such a structure requires some incubation time. Further extension of the time causes an increase in its thickness (Figure 5c,d) and an increase in the Fe_2_Al_5_ phase layer.

## 4. Conclusions

The effect of Mg additions on the formation of the periodic layered structure in a Zn-AlMg bath was investigated and following conclusions were obtained.

(1)The periodic layered structure is made of alternating FeAl_3_ phase layers and a bath alloy (Zn + Al + Mg). The alloy layers (Zn + Al + Mg) are composed of dendrites of a solid solution of Al in Zn. The interdendritic spaces are filled mainly by the MgZn_2_ intermetallic phase where small areas of Al in Zn solution are also located.(2)Additives of 3 wt.% Mg for Zn-23Al and Zn-31Al baths reduce the thickness of the coating, while the addition of 6 wt.% Mg causes complete loss of the periodic layered structure in coatings obtained in the Zn-23Al bath and limitation of its growth in the Zn-31Al bath.(3)The limitation of the growth of the periodic layered structure in the coating obtained in the Zn-31Al6Mg bath is due to the formation of a compact layer on the substrate composed of an FeAl_3_ phase layer containing MgZn_2_ phase precipitates and Fe_2_Al_5_ phase layer. This structure does not tend to detach the FeAl_3_ phase sequentially.

## Figures and Tables

**Figure 1 materials-14-01259-f001:**
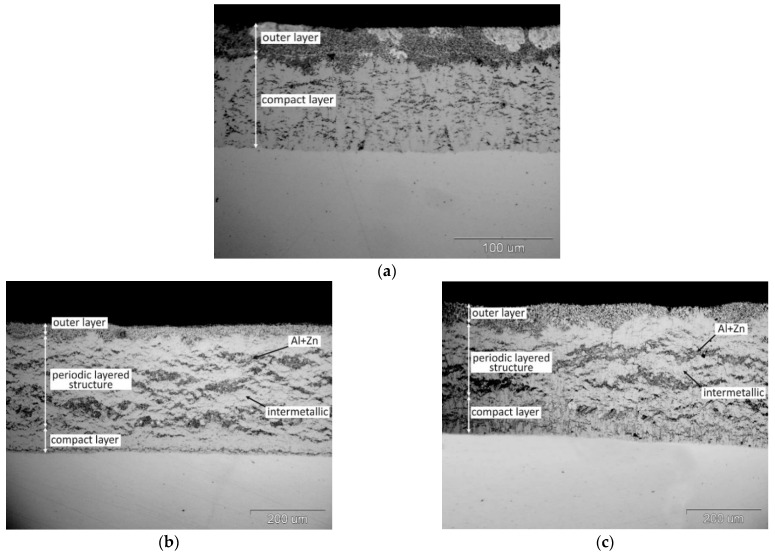
Structure of coatings obtained in two-component Zn-Al baths: (**a**) in Zn-15Al bath, air cooling; (**b**) in Zn-23Al bath, air cooling; (**c**) in Zn-23Al bath, water cooling; (**d**) in Zn-31Al bath, air cooling and; (**e**) Zn-31Al bath, water cooling; immersion time 120 s.

**Figure 2 materials-14-01259-f002:**
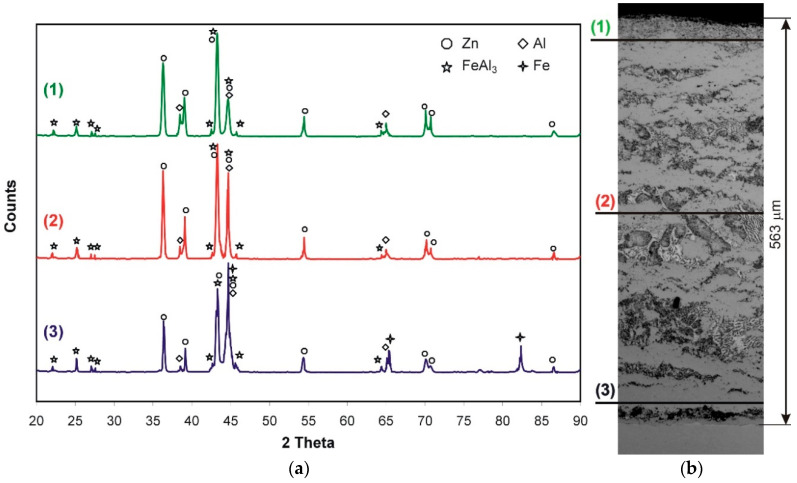
XRD of the coating obtained in Zn-23Al bath (**a**) and the structure with marked XRD test sections (**b**).

**Figure 3 materials-14-01259-f003:**
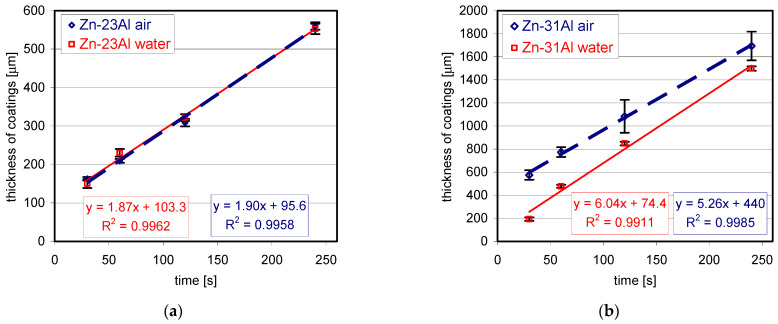
Growth kinetics of coatings obtained in the bath: (**a**) Zn-23Al and (**b**) Zn-31Al cooled in air and in water.

**Figure 4 materials-14-01259-f004:**
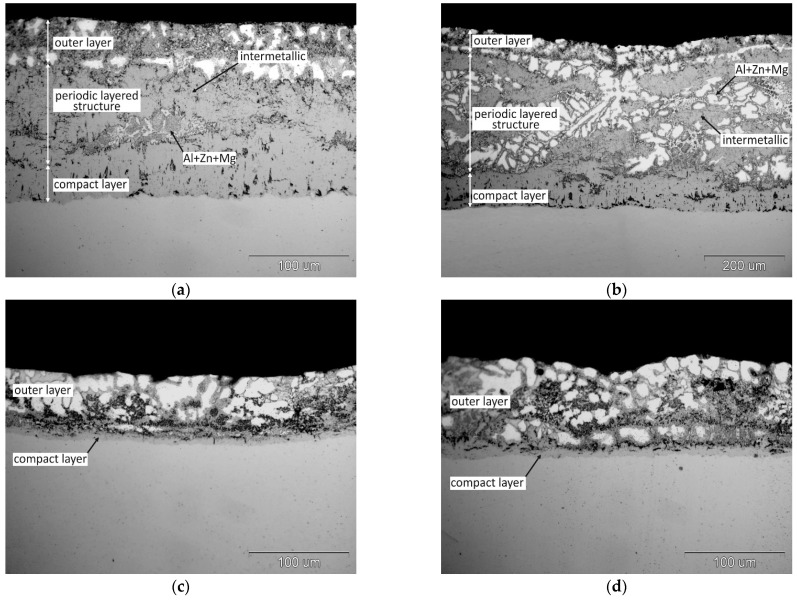
The structure of the coating obtained in the bath: (**a**) Zn-23Al3Mg at the immersion time of 120 s, (**b**) Zn-23Al3Mg at the immersion time of 240 s, (**c**) Zn-23Al6Mg at the immersion time of 120 s, (**d**) Zn-23Al6Mg at the immersion time of 240 s.

**Figure 5 materials-14-01259-f005:**
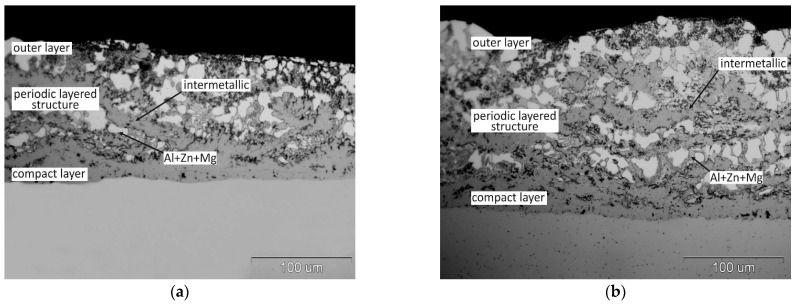
The structure of the coating obtained in the Zn-31Al6Mg bath during immersion: (**a**) 30 s, (**b**) 60 s, (**c**) 120 s, (**d**) 240 s.

**Figure 6 materials-14-01259-f006:**
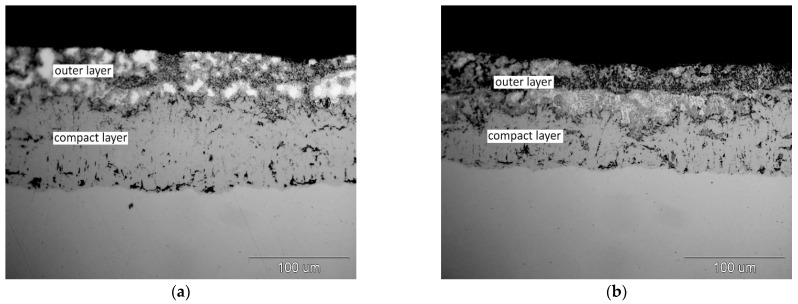
The structure of the coating obtained in the bath: (**a**) Zn-23Al with a 30 s immersion time (**b**) Zn-23Al3Mg with a 60 s immersion time.

**Figure 7 materials-14-01259-f007:**
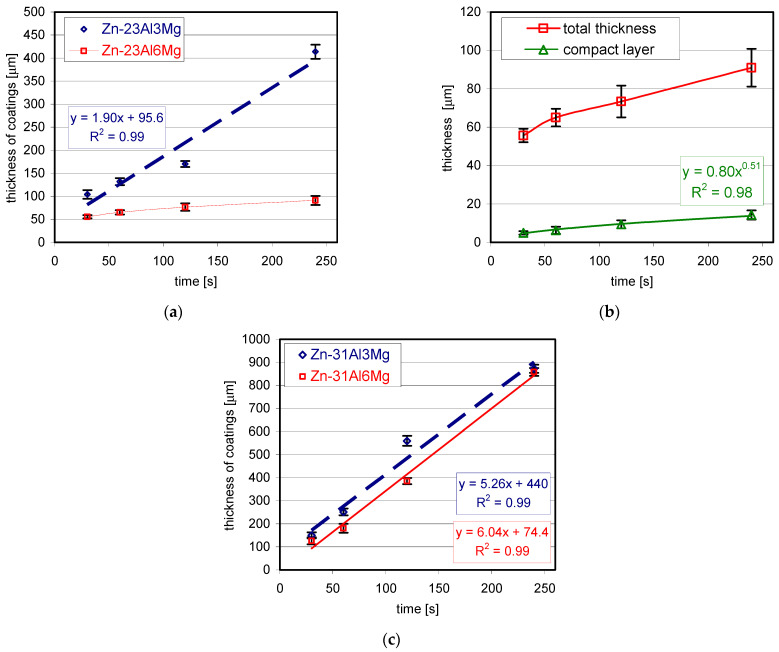
Growth kinetics of coatings obtained in the bath of (**a**) Zn-23Al and (**c**) Zn-31Al with the addition of 3 and 6 wt.% Mg and (**b**) total thickness and compact layer thickness of Zn-23Al6Mg coatings.

**Figure 8 materials-14-01259-f008:**
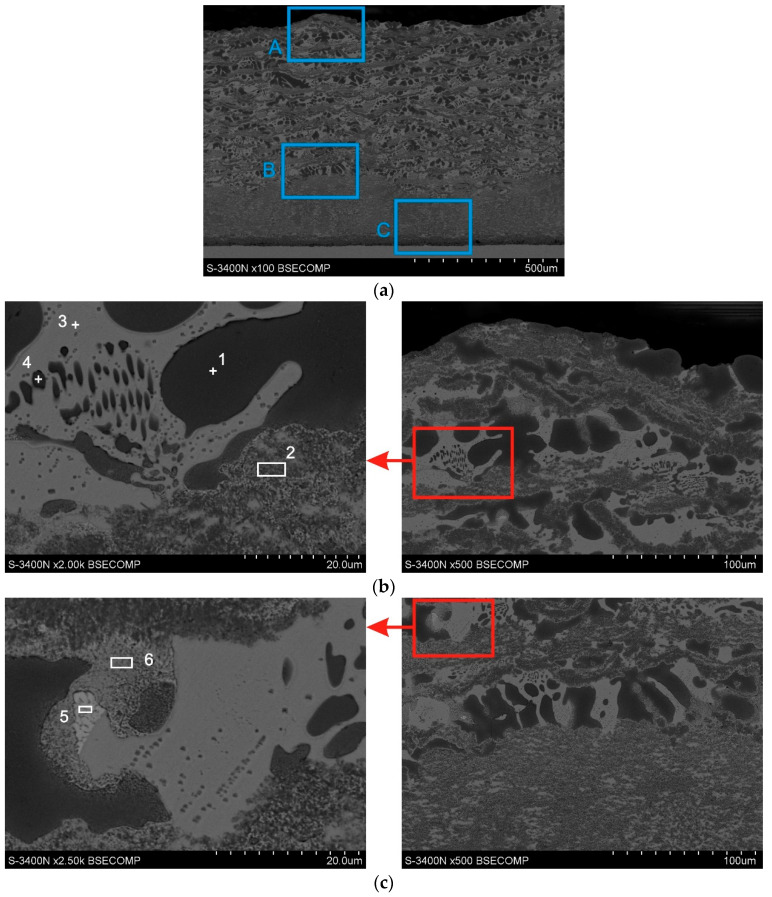
SEM images of Zn-31Al6Mg coatings obtained with an immersion time of 240 s: cross-section of coating (**a**), periodic layered structure (**b**), periodic layered structure–compact layer interface (**c**), compact layer (**d**).

**Figure 9 materials-14-01259-f009:**
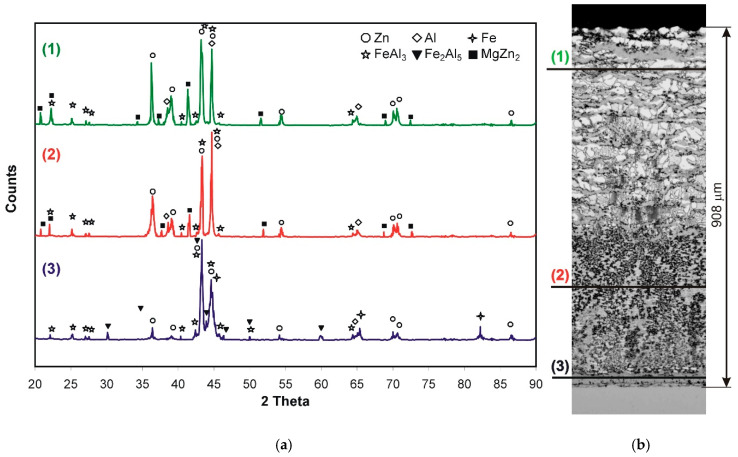
XRD of the coating obtained in Zn-31Al6Mg bath (**a**) and the structure with marked XRD test sections (**b**).

**Table 1 materials-14-01259-t001:** Element composition of Zn-31Al6Mg coating at points marked in Figure 8.

	Content of Elements							
	Mg-K		Al-K		Fe-K		Zn-K	
	wt.%	at.%	wt.%	at.%	wt.%	at.%	wt.%	at.%
1	-	-	71.0	85.6	-	-	29.0	14.4
2	1.1	1.9	37.5	56.9	25.8	18.9	35.6	22.3
3	13.9	29.5	2.4	4.5	-	-	83.7	66.0
4	2.9	4.2	56.2	73.7	-	-	40.9	22.1
5	1.3	3.3	3.0	6.9	-	-	95.7	89.8
6	-	-	22.9	41.9	-	-	77.1	58.1
7	-	-	47.1	66.1	32.6	22.1	20.4	11.8
8	12.6	27.3	2.0	3.4	2.3	1.9	95.7	67.4
9	1.1	2.2	24.8	42.3	20.3	16.8	54.9	38.7
10	-	-	54.9	71.6	45.1	28.4	-	-

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
