# Peer review of "Effect of Mg on the Formation of Periodic Layered Structure during Double Batch Hot Dip Process in Zn-Al Bath"

_materials, 2021, doi:10.3390/ma14051259_

Round 1

Reviewer 1 Report

The word "mortar" is not correct, it should be "alloy"

References in some cases do not correlate with the text, for example Liu et al [37]. Please correct the reference numbers in the text with the reference list

Author Response

Dear Reviewer,

We are grateful for taking your time to read our paper and for their constructive comments. We have carefully reviewed the comments and have revised the manuscript accordingly. Our responses are below given in a point-by-point manner. Changes to the text (manuscript) are shown in green in the revised manuscript. We hope the revised version is now suitable for publication.

The word "mortar" is not correct, it should be "alloy"

According to the reviewer's comment, the word "mortar" has been changed to "alloy" – line 74:

Mg was 3 and 6 wt. %. The charge materials for the bath were Super High Grade Zinc, ZnAl4 alloy and AlMg25 alloy. After removing from the Zn-Al bath, the samples were ..

References in some cases do not correlate with the text, for example Liu et al [37]. Please correct the reference numbers in the text with the reference list

Revision of the references in the text was done. The following changes have been made:

- In line 80 the error was corrected by changing references [46] to [19]

(…… of Zn-Al and Zn-AlMg alloys [19], it…)

- In line 140, 144, 153, 395, 406, et.al was added

- In line 383 the error was corrected by changing references [37] to [13]. (al. [13] claims that the MgZn2)

Reviewer 2 Report

The manuscript investigates the microstructure of ZnAl and ZnAlMg coatings obtained by hot dip galvanizing method on steel. The authors have deposited a substantial number of coatings with different compositions or deposition parameters and characterized their composition and structure. Although the paper provides quite some overview of the coating structure depending on composition, provided analysis lacks clarity and scientific soundness. The reviewer has also concerns regarding the interpretation of provided experimental results. It is unclear what new scientific or technological insights this paper provides. Also missing is a link between the described structure and properties of the coatings.

Some concerns in more details are provided below.

  1. As a reader I do not really see “periodic layered structures” which are frequently mentioned in the manuscript. In particular, in Figs 1 and 4 the regions marked as “periodic layered structures” are neither periodic nor layered, or at least not more than any mixture of different phases. The SEM analysis provided in Fig. 8 (which is, as I understand is the same material as in Fig. 5d) shows that the microstructure is neither periodic nor layered. So why calling it so? Why “periodic layered structure” is important? What properties does it bring?
  2. The same question regarding the “compact layer”: how the authors decide where it starts and ends? Why it is called compact? Are all other layers not “compact”, i.e. are porous? In Figs. 1b, 1c, 4a, 5c, 5d it is rather difficult to understand how the dimensions of “compact layer” were deduced.
  3. The XRD spectra in Fig. 2 seem to show no difference between lines 1, 2, and 3. The line “3” demonstrate additional peaks attributed to “Fe” but it comes most probably from the substrate. Generally, XRD does not detect elements but the distances between atomic planes. Which phase of “Fe” is shown there and where are the substrate peaks?
  4. The EDX results (Table 1) are poorly linked to the XRD characterization. The authors should better indicate how the phases they detected via XRD are correlated with provided EDX data. The XRD spectra contain peaks attributed to pure Zn and pure Al, where are those areas in EDX?

In general the manuscript sounds more like a project report listing all the work which was done for some experts in a specific scientific area. For a non-expert in hot galvanizing it is unclear how provided results expand the understanding of structure-property relationship or which important scientific or technological conclusions can be drawn.

It is suggested to make the manuscript more concise by significantly reducing its length, showing only the most important information and having a common thread throughout the manuscript. More importantly, scientific/technological value of provided results should be highlighted. What new has been done? What interesting/new/unexpected phenomena are observed? What new can be learned? And finally, how provided conclusion can help to understand the process better or to improve coating characteristics? Of course, additional information about anti-corrosion performance of considered coatings would significantly improve the manuscript.

Author Response

Dear Reviewer,

We are grateful for taking your time to read our paper and for their constructive comments. We have carefully reviewed the comments and have revised the manuscript accordingly. Our responses are below given in a point-by-point manner. Changes to the text (manuscript) are shown in green in the revised manuscript. We hope the revised version is now suitable for publication.

It is unclear what new scientific or technological insights this paper provides. Also missing is a link between the described structure and properties of the coatings.

In general the manuscript sounds more like a project report listing all the work which was done for some experts in a specific scientific area. For a non-expert in hot galvanizing it is unclear how provided results expand the understanding of structure-property relationship or which important scientific or technological conclusions can be drawn.

It is suggested to make the manuscript more concise by significantly reducing its length, showing only the most important information and having a common thread throughout the manuscript. More importantly, scientific/technological value of provided results should be highlighted. What new has been done? What interesting/new/unexpected phenomena are observed? What new can be learned? And finally, how provided conclusion can help to understand the process better or to improve coating characteristics? Of course, additional information about anti-corrosion performance of considered coatings would significantly improve the manuscript.

Zn-Al and Zn-Al-Mg coatings have been produced on sheets using the continuous method (Sendzimir method) for many years. The conducted research and many years of exploitation of these coatings under the conditions of use confirmed their very high corrosion resistance, several times higher than traditional zinc coatings. However, these coatings are not produced on an industrial scale by batch hot dip galvanizing. This is due to technological difficulties, mainly the necessity to use fluxing in this method.

Due to the very good properties of coatings on metal sheets, research has been conducted for years on the possibility of producing Zn-Al coatings, and in the last dozen or so years also Zn-Al-Mg coatings by batch hot dip galvanizing methods, as evidenced by numerous publications on this subject. Fluxing is not the only obstacle in the production of these coatings. Reactions between Fe and the Zn-Al and Zn-Al-Mg baths are very fast and lead to the formation of very thick coatings, which also results in the formation of a periodic layered structure. In their research, many authors produce coatings in new fluxes while  using very high temperatures (590-610oC). The authors of the article emphasize this in lines 236-241. High temperature causes excessive dissolution of iron, which significantly limits the applicability of such a process on an industrial scale. In the article, coatings were produced by the two-stage immersion method, which allowed to reduce the bath temperature to the lowest possible value, taking into account the technological possibilities of carrying out the process with a given bath composition. The choice of bath temperature is explained in the article in lines 76-82.

In the opinion of the authors, these technological problems should not raise doubts for experts. However, we agree with the reviewer's comment that they may be unclear to non-experts. Therefore, the text in lines: 57-63 was supplemented.

Due to the difficulty of producing continuous coatings in Zn-Al-Mg baths by batch hot dip methods, so far research on shaping the coating structure has been carried out at very high temperatures (590-610oC) [10, 20, 28]. High bath temperatures increase the tendency to form a periodic layered structure. The use of the double batch hot dip method for the production of coatings allows to lower the bath temperature while ensuring the processability and continuity of the obtained coatings. The study investigated the effect of Mg addition to the Zn-Al baths on the formation of periodic layered structure in coatings obtained by double batch hot dip methods.

Referring to the research on the properties of coatings, the main aim of the research presented in the article was to determine the effect of Mg addition to the Zn-Al baths on structure formation, including the periodic layered structure that is formed in the binary Zn-Al baths. For this purpose, a full range of studies of the ZnAl and Zn-Al-Mg bath compositions and process parameters were carried out. The article presents the results of the research that allowed to document the achieved effect. The formation of excessively thick coatings is a serious problem that is relatively often observed in the hot-dip galvanizing process. This is especially true of Sandelin's steel and Sebisty's steel. The formation of a periodic layered structure leads to the production of excessively thick coatings. On the other hand, the paper shows that the addition of 6 wt.% Mg inhibits the development of such a structure in baths with 23 wt.% Al and enables the change of the coating growth kinetics from linear to parabolic. This allows easy control of the coating thickness. In the Zn-31Al6Mg bath, stages of Mg influence on the structure change were observed, therefore detailed studies of this bath were presented. We agree with the reviewer's comment that corrosion resistance is a very important property of coatings. However, in our opinion, it is not advisable to test the periodic layered structure corrosion resistance, as it is not a structure at which the thickness of the coating can be easily controlled. What is important, however, is the corrosion resistance of coatings in which the formation of periodic layered structure has been limited. Due to the extensive content of this article, the corrosion resistance of these coatings will be presented in the next article.

As a reader I do not really see “periodic layered structures” which are frequently mentioned in the manuscript. In particular, in Figs 1 and 4 the regions marked as “periodic layered structures” are neither periodic nor layered, or at least not more than any mixture of different phases. The SEM analysis provided in Fig. 8 (which is, as I understand is the same material as in Fig. 5d) shows that the microstructure is neither periodic nor layered. So why calling it so? Why “periodic layered structure” is important? What properties does it bring?

Coatings obtained by the hot dip galvanizing process are formed in two stages. During immersion in the bath, a diffusion layer is built up, which consists of intermetallic phases of the reacting components. This layer becomes covered with a layer of bath melt when the product is withdrawn from the bath.

The layering of intermetallic phases in coatings obtained by a hot dip process is very complex. The growth of the intermetallic phase layer consists of simultaneous partial processes: the reaction diffusion of the components, dissolution at the interface of solid phase/bath and secondary crystallization from the bath supersaturated with iron. If the hot dip galvanizing process takes place in a Zn bath, compact layers of the Fe-Zn intermetallic phases are usually formed. The phenomenon of delamination of the intermetallic phase layer during immersion in the bath is well known in the Zn-Al and Al baths. After delamination, as a result of direct contact of the bath with the substrate, the layer of intermetallic phase grows again, which increases until delamination again. Until detachment, the layer structure is compact. It is a cyclical process leading to the formation of alternating layers of intermetallic phase and bath melt, which form a periodic structure. These layers are highlighted and marked in the figures as intermetallic and (Zn+Al) or (Zn+Al+Mg). It should also be added that the hypothesis of detachment, or the so-called: spalling effect is quoted in lines 388-395 and they explain the delamination mechanism by the occurrence of stresses in the layer, which can lead to the formation of fragments of intermetallic phase layers as shown in the figures.

The SEM in Fig. 8 shows at high magnification the repeating element of the periodic layered structure, i.e. the FeAl3 phase zone and the bath zone (Zn+Al+Mg). On the other hand, the identified structural components of MgZn2 as well as α and β solid solutions are part of the bath alloy structure (Zn+Al+Mg).

From a technological point of view, it is disadvantageous to create a periodic layered structure which leads to an excessively thick coating. Therefore, the study of the properties of this layer, mentioned by the reviewer, does not seem justified. Moreover, the aim of the article is to show the influence of Mg on limiting the growth of the periodic layer structure.

The same question regarding the “compact layer”: how the authors decide where it starts and ends? Why it is called compact? Are all other layers not “compact”, i.e. are porous? In Figs. 1b, 1c, 4a, 5c, 5d it is rather difficult to understand how the dimensions of “compact layer” were deduced.

The boundaries of the coating zones were determined on the basis of the assessment of the coating morphology on its cross-section. If the FeAl3 intermetallic phase layer does not show continuity in the cross-section, it means that it is not compact. The FeAl3 intermetallic phase layers alternate with the bath alloy layers (Zn+Al+Mg). The FeAl3 phase itself is not compact, but it also does not mean that the coating is porous. At the same time, it should be noted that a similar method of describing layers in the structure of coatings obtained by the hot dip method has been used for a long time. For example, the δ1 phase layer (present in hot dip galvanizing coatings) can be given, which very often in scientific descriptions (including in some cases on the Fe-Zn phase equilibrium system) is differentiated morphologically as two zones: δ1c (compact) and δ1p (palisade).

The description of the structural elements used in the manuscript is intended to facilitate the correlation of the text with the images of the structure.

The XRD spectra in Fig. 2 seem to show no difference between lines 1, 2, and 3. The line “3” demonstrate additional peaks attributed to “Fe” but it comes most probably from the substrate. Generally, XRD does not detect elements but the distances between atomic planes. Which phase of “Fe” is shown there and where are the substrate peaks?

According to the cited structure image in Fig. 3b on lines 1 and 2, no differences in the morphology of the coating are observed, so the diffraction patterns indicate the same phases. Iron is detected on line 3. The cross-section on line 3 is simultaneously adjacent to the boundary of the coating with the substrate. We agree with the reviewer's comment that Fe is a component of the substrate here. Therefore, the statement that the presence of Fe indicates that the measurement area encompassed the coating/substrate boundary was given in lines: 136-138.

We agree with the reviewer's comment that XRD studies do actually determine the distances between atomic planes. However, specifying the crystal lattice parameter makes it difficult to read the diffractogram, therefore the identified Al, Zn and FeAl3 phases (which should be interpreted with the EDS results) and Fe, which is the main component of the steel, were marked.

The EDX results (Table 1) are poorly linked to the XRD characterization. The authors should better indicate how the phases they detected via XRD are correlated with provided EDX data. The XRD spectra contain peaks attributed to pure Zn and pure Al, where are those areas in EDX?

XRD analysis allows to identify the crystal lattice corresponding to metals: Al and Zn. It does not allow the identification of solutions of these metals. According to EDS studies, Al-rich regions contain Zn, while Zn-rich regions contain Al. We agree with the Reviewer's remark that the interpretation of the XRD results should refer in this respect to the EDS results. Therefore, in lines: 346-354, the results of SEM-EDS and XRD tests were summarized and correlated, indicating the phases and their location in the coating, taking into account the presence of solid solutions of Al in Zn and Zn in Al.

Reviewer 3 Report

Questions or recommendation:

Rows 66, 67 – it is not known expression “mortar”.

Row 77, in HDG pickling in pretreatment process is used instead etching.

It would be useful to explain at the beginning of chapter 3 what is transition layer, compact layer and outer layer.

What is the proof of the distance reached after grinding? (Row 93)

Fig. 1. Structure of coatings in two-component Zn-Al baths:  “there is no stated immersion time”.

Row 128: …in the Al layer formed on the iron surface….. –“what Al layer???”

Row 200: (Zn+al+Mg) – Al not al

Statement on rows 218-219 does not match the microstructure on Fig. 5.  That correspondents only for Zn-23Al6Mg immersion time 120s and 240s (Fig.4).

Row 245: Fig. 5d is not for immersion time 120s, but for 240 s.

Rows 303-304:……presence of Al solution in Zn?? Presence Al in Zn solution???

Rows 329-330:  …and Fe intermetallic phase was confirmed.   “What type?”

Rows 363-364: 15-55wt% of what and 0,5-3,0wt% of what???

Row 374: Fe2Al5Znx - subscripts

Author Response

Dear Reviewer,

We are grateful for taking your time to read our paper and for their constructive comments. We have carefully reviewed the comments and have revised the manuscript accordingly. Our responses are below given in a point-by-point manner. Changes to the text (manuscript) are shown in green in the revised manuscript. We hope the revised version is now suitable for publication.

Questions or recommendation:

Rows 66, 67 – it is not known expression “mortar”.

According to the reviewer's comment, the word "mortar" has been changed to "alloy"

Row 77, in HDG pickling in pretreatment process is used instead etching.

The preparation of the products surface before galvanizing includes "etching" in HCl solution. This operation is commonly called "pickling" both in technological documentation and in scientific studies. We therefore suggest that "pickling" should be left in the manuscript although we agree with the Reviewer that the use of this word is not fully understood.

It would be useful to explain at the beginning of chapter 3 what is transition layer, compact layer and outer layer.

The text was completed in line: 108-113

A compact layer is formed at the substrate, which forms the transition layer of the coating. The layer increases with immersion in the bath. The compact layer has a considerable thickness and its morphology is typical for the phases of the Fe-Al system growing on a steel substrate in Zn-Al baths [20]. The compact layer is covered with an outer layer, which is formed as a result of pulling the Zn-Al bath alloy and its crystallization on the sample surface [21].

What is the proof of the distance reached after grinding? (Row 93)

After grinding, a direct observation was made on the coating cross-section. Observations on the light microscope allowed to identify the boundaries of the coating after grinding and determining the depth of grinding.

To clarify these doubts, the text in lines 100-102 was added to the manuscript:

The depth of grinding the coating was determined by observing the cross-section of the coating under a light microscope.

Fig. 1. Structure of coatings in two-component Zn-Al baths: “there is no stated immersion time”.

The immersion time has been completed in the description of Fig. 1 - line 117:

Figure 1. Structure of coatings obtained in two-component Zn-Al baths: (a) in Zn-15Al bath, air cooling, (b) in Zn-23Al bath, air cooling, (c) in Zn-23Al bath, water cooling, (d) in Zn-31Al bath, air cooling and (e) Zn-31Al bath, water cooling; immersion time 120 s.

Row 128: …in the Al layer formed on the iron surface….. –“what Al layer???”

The error in the sentence (line 128 in manuscript, 138 in revised manuscript) has been corrected.

Many authors argue that in the layer formed on the iron surface in the Zn bath, containing Al, the Fe2Al5 phase is the major component of the transition layer.

Row 200: (Zn+al+Mg) – Al not al

The error has been corrected.

Line 212:

layer (Zn+Al+Mg) are formed during immersion of 120 and 240 s (Fig. 4c, d).

Statement on rows 218-219 does not match the microstructure on Fig. 5. That correspondents only for Zn-23Al6Mg immersion time 120s and 240s (Fig.4).

The text has been changed to make it easier to understand.

Line 230-232:

Based on the presented research results, it can be assumed that with the immersion time of 120 and 240 s, the formation of periodic layered structure disappears in the Zn-23Al6Mg bath, while its significant limitation is observed in the Zn-31Al6Mg bath.

Row 245: Fig. 5d is not for immersion time 120s, but for 240 s.

The text has been changed, now everything is ok.

Line 256-259:

Also in the Zn-23Al3Mg bath, the initiation of periodic layered structure formation observed after 120s (Fig. 4a) confirms that its formation requires an incubation time, which depends on the bath composition and process temperature.

Rows 303-304:……presence of Al solution in Zn?? Presence Al in Zn solution???

A fair point, of course. The error has been corrected.

Line 318:

the presence of Al solution in Zn.

Rows 329-330:  …and Fe intermetallic phase was confirmed.   “What type?”

A fair point, of course. The error has been corrected.

Line 343-344:

presence of intermetallic phase Fe2Al5 and Fe

Rows 363-364: 15-55wt% of what and 0,5-3,0wt% of what???

Elements have been supplemented in the text.

Line 378:

in baths with a content of 15-55 wt.% Al and 0.5-3.0 wt.% Mg, claims that Mg does not

Row 374: Fe2Al5Znx - subscripts

The error was corrected.

Line 389:

…., the Fe2Al5Znx phase is
